# Synthesis of Pluri-Functional Amine Hardeners from Bio-Based Aromatic Aldehydes for Epoxy Amine Thermosets

**DOI:** 10.3390/molecules24183285

**Published:** 2019-09-09

**Authors:** Anne-Sophie Mora, Russell Tayouo, Bernard Boutevin, Ghislain David, Sylvain Caillol

**Affiliations:** 1Institut Charles Gerhardt, UMR 5253—CNRS, Université de Montpellier, Ecole Nationale Supérieure de Chimie de Montpellier, 240 Avenue Emile Jeanbrau, 34296 Montpellier, France; 2SAS Nouvelle Sogatra, 784 Chemin de la Caladette, 30350 Lezan, France

**Keywords:** imine, epoxide, amine, thermoset, bio-based

## Abstract

Most of the current amine hardeners are petro-sourced and only a few studies have focused on the research of bio-based substitutes. Hence, in an eco-friendly context, our team proposed the design of bio-based amine monomers with aromatic structures. This work described the use of the reductive amination with imine intermediate in order to obtain bio-based pluri-functional amines exhibiting low viscosity. The effect of the nature of initial aldehyde reactant on the hardener properties was studied, as well as the reaction conditions. Then, these pluri-functional amines were added to petro-sourced (diglycidyl ether of bisphenol A, DGEBA) or bio-based (diglycidyl ether of vanillin alcohol, DGEVA) epoxy monomers to form thermosets by step growth polymerization. Due to their low viscosity, the epoxy-amine mixtures were easily homogenized and cured more rapidly compared to the use of more viscous hardeners (<0.6 Pa s at 22 °C). After curing, the thermo-mechanical properties of the epoxy thermosets were determined and compared. The isophthalatetetramine (IPTA) hardener, with a higher number of amine active H, led to thermosets with higher thermo-mechanical properties (glass transition temperatures (*T_g_* and *T_α_*) were around 95 °C for DGEBA-based thermosets against 60 °C for DGEVA-based thermosets) than materials from benzylamine (BDA) or furfurylamine (FDA) that contained less active hydrogens (*T_g_* and *T_α_* around 77 °C for DGEBA-based thermosets and *T_g_* and *T_α_* around 45 °C for DGEVA-based thermosets). By comparing to industrial hardener references, IPTA possesses six active hydrogens which obtain high cross-linked systems, similar to industrial references, and longer molecular length due to the presence of two alkyl chains, leading respectively to high mechanical strength with lower *T_g_*.

## 1. Introduction

Amine is one of the most important functional groups in the chemical industry, highly present in various industrial fields, such as pharmaceuticals [1,2,3], agrochemicals [4,5], detergents [6,7], lubricants [8] and polymer industry [9,10]. Amines act as intermediates in the synthesis of different polymers including phenolic resins [11,12], polyimides [13], polyureas, polyurethane [14] and poly(hydroxy)urethanes [15], polyamide [16,17] and epoxy thermosets [18,19]. Our team has a long experience in the synthesis of epoxy thermosets [10,20] and has recently developed new efficient bio-based amine hardeners exhibiting high reactivity, by a direct amination method of epoxy monomers using an aqueous ammonia solution [21]. Due to the presence of many hydrogen bond sites on their structures, the obtained β-hydroxylamine hardeners showed high reactivity but also high viscosity. Therefore, we worked on an alternative pathway to synthesize bio-based pluri-functional amine monomers, avoiding the formation of hydroxyl functions, in order to disfavor the hydrogen bonds effect and thus decrease viscosity. Hence, we synthesized amines from imine reduction, more generally called reductive amination. This methodology allows easily obtaining secondary or tertiary amine monomers with high yields and high reactivity [22,23,24]. Using aromatic aldehydes, amines monomers containing aromatic moieties can be synthesized from aliphatic amines which are less toxic than conventional aromatic amines. Moreover, benzyl amine monomers are much more reactive than aromatic ones. Aromaticity is really interesting in epoxy-amine formulation to improve the miscibility of amines with epoxy monomers, which are most of the time aromatic substances. Moreover, providing aromaticity tunes the hydrogen equivalent weight (HEW), decreases the volatility of amines and induces high thermo-mechanical properties for the final thermosets. Hence, such aromatic amines could be very highly desirable for epoxy thermosets synthesis.

Moreover, the imine reduction is a really simple method, using a variety of hydride source reducing agents [25,26]. The reduction step can also be performed by the aminocatalytic method using a catalyst [27], by one-step bio-catalyzed transformation using enzymes [28,29,30] or by the adaptation of the Haber–Bosch method used for the synthesis of ammonia [31]. This method involves the metal-catalyzed hydrogenation of unsaturated bonds as in the case of imine functions [32,33,34,35,36]. For instance, Jia et al. described the development of the ruthenium-catalyzed hydrogenation method for the reduction of imine intermediates to obtain amine monomers [37]. Exposito et al. recently developed the imine Pd-catalyzed hydrogenation using the industrial continuous-flow process, thanks to a Pd/C support, which was stable over multiple reaction–regeneration cycles [38].

For epoxy-amine thermosets formulation, the hardener requires a structure with at least three active hydrogens to form a cross-linked network when using a conventional diepoxy prepolymer, such as diglycidyl ether of bisphenol A (DGEBA). Most of the amine hardeners are petro-based diamines (four active H) such as 4,4′-Methylenebis(cyclohexylamine) (PACM) [39,40] and Triethyleneglycol diamine (EDR 148) [41,42], and there are only a handful of bio-based diamines, such as 1,6-hexamethylenediamine or 1,9-nonanediamine which can be obtained from renewable resources [43], 1,10-decanediamine from alcohols [44] and the diamine developed by Garrison et al. from renewable terpenoids [45]. However, beyond this functionality, very few polyfunctional amines are reported, and most of the time they are petro-based [10], such as Jeffamine T403, which exhibits a low reactivity [46]. Only a few poly-functional amine hardeners are bio-based, such as citric amido amine described by Bähr et al., which performed the aminolysis of citric acid to obtain a tri-functional primary amine [47]. Pluri-functional cardanol-based amines can also be obtained with formaldehyde and aliphatic di- or triamines to obtain the corresponding Mannich hardeners [48].

The reductive amination could be a solution to design polyfunctional amine hardeners since this is an easy method to tune and modify already existing amines. However, the imine reduction generally leads to secondary mono- or diamines as intermediates in pharmaceutical applications [49,50,51], except when using ammonia as the initial amine reactant [52,53], reaching primary amines which can be efficient hardeners [54,55]. This process was used for producing diamine monomers, which are useful as starting materials for various polyamides and polyurethanes but not for epoxy thermosets. Moreover, only a few amine monomers from this reductive amination method are bio-based such as 2,5-bis(aminomethyl)furan developed in 2015 by Le et al. [10,56,57]. Thanks to reductive amination, another possibility to synthesize hardeners is to slowly add an aldehyde in an excess of primary diamine monomers. Only a few examples are described in the literature. For instance, Micklitsch et al. synthesized an amine monomer exhibiting one secondary and one primary amine functions from ethylenediamine and benzaldehyde in the presence of NaBH_4_ [58]. In 2012, Milelli et al. described the synthesis of a naphthalene diimide derivative, obtained from the addition of reduced benzylic imine on 1,4,5,8-naphthalenetetracarboxylic dianhydride monomer [59]. To the best of our knowledge, the reductive amination, thanks to imine formation, is hardly described in the literature for epoxy-amine applications. Only Kasemi et al. patented hardeners using aldehyde monomers and principally *m*-xylylenediamine (MXDA) or some derivatives of 1,2-propylenediamine as amine without any concerns about the biomass origin and reactant toxicity [60]. Moreover, this patent is very broad and imprecise and does not study the properties of hardeners.

Hence, our work proposes for the first time a reductive amination method for the synthesis of pluri-functional amine hardeners containing aromatic moieties, between three to six active H, forming a cross-linked system using diepoxy monomers. Moreover, this method respects green chemistry concepts due to the use of monomers that are or could be obtained from biomass and with a low toxicity. We have studied the impact of reaction conditions on the final amine functionality. Then, we have evaluated the impact of phenyl and furan moieties on the thermal resistance of monomers [61,62]. Finally, hardener properties were studied with the synthesis and the characterization of epoxy-amine thermosets.

## 2. Experimental

### 2.1. Materials and Methods

Benzaldehyde (purity ≥ 99.0%), 1,5-diamino-2-methylpentane as DYTEK^®^A (purity ≥ 99.0%), furfural (purity ≥ 99.0%), 1,6-hexanediamine (purity ≥ 98.0%) and terephthaldehyde (purity ≥ 99.0%) were purchased from Sigma-Aldrich (St. Quentin Fallavier, France). 2-methyltetrahydrofuran (purity ≥ 99.0%) was purchased from Alfa Aesar (Kandel, Germany). Isophthalaldehyde (purity ≥ 98.0%) was purchased from TCI Europe NV (Zwijndrecht, Belgium). Ethyl acetate (purity ≥ 99.9%) was purchased from VWR Chemicals (Fontenay-sous-Bois, France). Deuterated solvents were obtained from Sigma Aldrich for NMR study.

### 2.2. Characterization Techniques

^1^H and ^13^C NMR analyses were recorded on a 400 MHz Bruker Aspect NMR spectrometer at 23 °C (Rheinstetten, Germany) in deuterated solvents. Tetramethylsilane was used as a reference to the chemical shifts, which are given in parts per million (ppm).

Viscosities measurements were performed at 22 °C on the AR-1000 rheometer (TA Instruments, New Castle, DE, USA). A 60 mm diameter and 2° cone-plan geometry was used. The flow mode was used with a gradient from 1 to 0.01 rad⋅s^−1^.

Fourier-transform infrared (FTIR) spectra were recorded using a Thermo Scientific Nicolet 6700 FTIR spectrometer with “diamond ATR” equipment (Waltham, MA, USA) in transmittance and with a band accuracy of 4 cm^−1^. Additionally, 32 scans were performed in the range of 4000–650 cm^−1^ and with a resolution of 4. OMNIC software was used.

Thermogravimetric analyses (TGA) were recorded using a Netzsch F1-Libra analyzer (Selb, Germany) at a heating rate of 20 °C⋅min^−1^ from 25 to 600 °C (nitrogen stream). Each sample was placed in an alumina crucible and contained an amount between 9–10 mg. The moisture and volatile content, the percentage of residue at 600 °C, and the degradation temperature (*T_d_*) were determined thanks to TGA analyses.

Differential scanning calorimetry (DSC) measurements were performed with the use of a Netzsch DSC200F3 calorimeter F3 (Selb, Germany, indium calibration, nitrogen stream). Pierced aluminum pans were used as crucibles with approximately 10 mg of the sample. A heating rate of 20 °C⋅min^−1^ from −100 °C to 120 °C was used to record the glass transition temperature (*T_g_*). Right values were measured on a second heating ramp.

Dynamic mechanical analyses (DMA) were performed on Metravib DMA 25 with Dynatest 6.8 software (TA Instruments, New Castle, DE, USA). Uniaxial stretching of samples was carried out while heating at a rate of 3 °C⋅min^−1^ using a constant frequency of 1 Hz and a fixed strain (according to sample elastic domain). For *T_g_* around 60 °C, DMA analyses were performed from –30 °C to + 160 °C. For *T_g_* around 90 °C, DMA analyses were performed from 0 °C to +190 °C.

Cross-linking density: From rubber elasticity theory [63], uniaxial stretching was studied on the rubbery plateau at T = *T_α_* +80 °C, and at very small deformations. According to these hypotheses, the cross-linking density (ν′), was determined from Equation (1), where *E*′ is the storage modulus, R is the universal gas constant and *T_α_* is the vitreous transition temperature, in K. Calculated values are given for informational purposes only, and they can only be compared.
(1)$$\nu^{\prime }=\frac{E_{T_{\alpha +80}}^{\prime }}{3RT_{\alpha +80}}$$

Swelling indices (SI) were measured with a sample of approximately 25 mg, which was placed in 25 mL of tetrahydrofuran (THF) for 24 h. This step was repeat three times for repeatability. The swelling index was calculated according to Equation (2), where *m*_1_ is the mass of the material after swelling in THF during 24 h and *m*_2_ is the initial mass of the material.
(2)$$SI=\frac{m_{1}-m_{2}}{m_{2}}\times 100$$

Gel contents (GC) were measured after SI samples were dried in a ventilated oven at 70 °C for 24 h. The gel content was calculated according to Equation (3), where *m*_3_ is the mass of the material after drying and *m*_2_ is the initial mass of the material.
(3)$$GC=\frac{m_{3}}{m_{2}}\times 100$$

### 2.3. Synthesis of Amine Hardeners

First, 100 mL of H_2_O-2-MeTHF mixture (70–30) were added to DYTEK^®^A (17.3 g, 149 mmol, 10 equivalents) in a 250 mL two-neck round-bottom flask. Then, isophthalaldehyde (2 g, 14.9 mmol, 1 equivalent) was solubilized in 30 mL of 2-MeTHF and then added dropwise using a dropping funnel. The reaction crude was stirred and heated until complete aldehyde conversion, at 110 °C. The solution was then cooled down to room temperature.

In the case of monoaldehyde use, only 5 equivalents of DYTEK^®^A were used for 1 equivalent of monoaldehyde.

In the case of one pot conditions for isophthalatetetramine (IPTA)2 synthesis:

Then, 100 mL of H_2_O-2-MeTHF mixture (70–30) were added to DYTEK^®^A (17.3 g, 149 mmol, 10 equivalents) in a 250 mL round-bottom flask. Then, isophthalaldehyde (2 g, 14.9 mmol, 1 equivalent), was added. The reaction crude was stirred and heated until complete aldehyde conversion, at 110 °C. The solution was then cooled down to room temperature.

For each case, 2 equivalents of sodium borohydride were then added slowly to the theoretical amount of imine and solvent mixture, and the reaction crude was heated at reflux temperature until complete disappearance of imine signal in ^1^H NMR. The solvent was removed under reduced pressure. The sodium borohydride was then neutralized by the addition of the reaction crude in water (200 mL). This aqueous solution was extracted with AcOEt (3 × 600 mL). Organic phase was washed with brine solution (400 mL). Then, organic phase was dried with MgSO_4_, filtered and the solvent was removed under reduced pressure. Any trace of DYTEK^®^A was removed by distillation.

All the described structures corresponded to the attack of each amine group from the diamine compound onto dialdehyde.

Isophtalatediiminediamine (IPDIDA) (C_20_H_34_N_4_) (Figure 1). Colorless liquid. IR (neat, ν, cm^−1^): 3381 and 3266 (N*_primary_*-H), 2923 and 2845 (C-sp^3^), 1643 (C = N), 1582 (*δ*N-H), 1458 (C = C*_aromatic_*), 1376, 1339, 1291, 1055, 1021, 967, 794. ^1^ H NMR (400 MHz, CDCl_3_, ppm) *δ*: 8.26 (s, 1H, CH_imine_), 8.23 (s, 1H, CH_imine_), 7.99 (m, 1H, H_Ar_), 7.75 (m, 2H, 2 × H_Ar_), 7.40 (m, 1H, H_Ar_), 3.58 (t, 2H, *J* = 6.0 Hz, HC = N-C*H*_2_-CH_2_), 3.46 (ddd as apparent dt, 2H, HC = N-C*H*_2_-CH), 2.66 (t, 2H, *J* = 6.5 Hz, H_2_N-C*H*_2_-CH_2_), 2.53 (ABX signal, 2 × dd, 2H, *J* = 12.6 and 6.9 Hz, 2 × H_2_N-C*H*a-CH, 2 × H_2_N-C*H*b-CH), 1.95–1.61 (m, 2H, 2 × CH), 1.55–1.07 (m, 12H, 4 × H_labiles_, 4 × CH_2_), 0.94–0.86 (m, 6H, 2 × CH_3_). ^13^ C NMR (101 MHz, CDCl_3_, ppm) *δ*: 160.5 and 160.4 (2 × CH_imine_), 136.7 (2 × Cq_Ar_), 129.8 and 129.7 (2 × CH_Ar_), 128.9 (CH_Ar_), 128.0 (CH_Ar_), 68.2 (HC-*C*H_2_-N = CH), 62.0 (H_2_C-*C*H_2_-N = CH), 48.4 (H_2_N-*C*H_2_-CH), 42.5 (H_2_N-*C*H_2_-CH_2_), 36.2 (CH), 34.2 (CH), 32.5–28.4 (4 × CH_2_), 18.2 and 17.5 (2 × CH_3_).

IPTA (C_20_H_38_N_4_) (Figure 2). Colorless liquid. IR (neat, ν, cm^−1^): 3278 (broad N-H), 2920 and 2849 (C-sp^3^), 1643–1454 (C = C*_aromatic_*), 1558 (*δ*N-H), 1376, 1291, 1155, 1003, 785. ^1^H NMR (400 MHz, CDCl_3_, ppm) *δ*: 7.26 (m, 4H, 4 × H_ar_), 3.80 (s, 2H, Cq_ar_-CH_2_), 3.79 (s, 2H, Cq_ar_-CH_2_), 2.71–2.43 (m, 8H, 4 × H_2_N-C*H*_2_), 2.01–1.10 (m, 16H, 6 × H_labiles_, 4 × CH_2_, 2 × CH), 0.94 (d, 3H, *J* = 6.7 Hz, CH_3_), 0.92 (d, 3H, *J* = 6.6 Hz, CH_3_). ^13^C NMR (101 MHz, CDCl_3_, ppm) *δ*: 140.6 and 140.5 (2 × Cq_Ar_), 128.4, 127.9, 127.8 and 126.7 (4 × CH_Ar_), 55.9 (HN-*C*H_2_-CH), 54.1 (2 × Cq_ar_-*C*H_2_), 49.9 (HN-*C*H_2_-CH_2_), 48.2 (H_2_N-*C*H_2_-CH), 42.2 (H_2_N-*C*H_2_-CH_2_).

Benzylimineamine (BIA) (C_13_H_20_N_2_) (Figure 3). Colorless liquid. IR (neat, ν, cm^−1^): around 3250 (N*_primary_*-H, which disappear during reversibility), 3061 and 3026 (C-sp^2^), 2926 and 2829 (C-sp^3^), 1643 (C = N), 1579 (*δ*N-H), 1493–1450 (C = C*_aromatic_*), 1309, 1293, 1218, 1169, 1074, 1025, 968, 848. ^1^H NMR (400 MHz, CDCl_3_, ppm) *δ*: 8.30 or 8.27 (s, 1H, CH_imine_), 7.76–7.71 (m, 2H, 2 × H_Ar_), 7.45–7.40 (m, 3H, 3 H_Ar_), 3.69–3.58 (m, 2H, HC = N-C*H*_2_-CH_2_) or 2.53 (ABX signal, 2 × dd, 2H, *J* = 11.4 and 7.1 Hz, HC = N-C*H*a-CH, HC = N-C*H*b-CH), 2.72 (t, 2H, *J* = 6.9 Hz, H_2_N-C*H*_2_-CH_2_) or 2.59 (ABX signal, 2 × dd, 2H, *J* = 12.6 and 6.0 Hz, H_2_N-C*H*a-CH, H_2_N-C*H*b-CH), 2.02–1.54 (m, 7H, 2 × H_labiles_, 2 × CH_2_, CH), 1.00–0.92 (m, 3H, CH_3_). ^13^C NMR (101 MHz, CDCl_3_, ppm) *δ*: 161.0 or 160.9 (CH_imine_), 136.4 (Cq_Ar_), 130.5 (CH_Ar_), 128.6–128.1 (4 × CH_Ar_), 68.2 (HC-*C*H_2_-N = CH) or 62.0 (H_2_C-*C*H_2_-N = CH), 48.4 (H_2_N-*C*H_2_-CH) or 42.6 (H_2_N-*C*H_2_-CH_2_), 34.3 (CH), 32.5–28.4 (2 × CH_2_), 18.2 (CH_3_). High Resolution Mass Spectrometry (HRMS) (*m*/*z*, ES+, [M + H^+^]): C_13_H_21_N_2_; calculated 205.1699; found 205.1706.

Benzyldiamine (BDA) (C_13_H_22_N_2_) (Figure 4). Colorless liquid. IR (neat, ν, cm^−1^): 3375–3287 (broad N-H), 3093, 3069 and 3026 (C-sp^2^), 2924 and 2850 (C-sp^3^), 1667–1452 (C = C*_aromatic_* and *δ*N-H), 1376, 1291, 1116, 1074, 1027, 806. ^1^H NMR (400 MHz, CDCl_3_, ppm) *δ*: 7.32–7.23 (m, 5H, 5 × H_ar_), 3.78 or 3.77 (s, 2H, Cq_ar_-CH_2_), 2.68–2.40 (m, 4H, 2 × H_2_N-C*H*_2_), 1.98–1.07 (m, 8H, 3 × H_labiles_, 2 × CH_2_, CH), 0.92 (d, 3H, *J* = 6.7 Hz, CH_3_) or 0.89 (d, 3H, *J* = 6.6 Hz, CH_3_). ^13^C NMR (101 MHz, CDCl_3_, ppm) *δ*: 140.6 or 140.4 (2 × Cq_Ar_), 128.4–126.9 (5 × CH_Ar_), 55.8 (HN-*C*H_2_-CH) or 49.7 (HN-*C*H_2_-CH_2_), 54.1 (Cq_ar_-*C*H_2_), 48.2 (H_2_N-*C*H_2_-CH) or 42.3 (H_2_N-*C*H_2_-CH_2_), 36.1 or 33.2 (CH), 32.0 (CH_2_), 27.5 (CH_2_), 18.1 or 17.4 (CH_3_). HRMS (*m*/*z*, ES+, [M + H^+^]): C_13_H_23_N_2_; calculated 207.1856; found 207.1862.

Furfurylimineamine (FIA) (C_11_H_18_N_2_O) (Figure 5). Brown liquid. IR (neat, ν, cm^−1^): around 3250 (N*_primary_*-H, which disappear during reversibility), 3110 (C-sp^2^), 2927 and 2871 (C-sp^3^), 1643 (C = N), 1579 (*δ*N-H), 1483–1459 (C = C*_aromatic_*), 1392, 1358, 1272, 1238, 1154, 1079, 1014, 931, 816. ^1^H NMR (400 MHz, CDCl_3_, ppm) *δ*: 8.30 or 8.27 (s, 1H, CH_imine_), 7.51 (d, 1H, *J* = 1.7 Hz, O-CH_Ar_), 6.72–6.71 (m, 1H, *H*C=CH-CH), 6.47 (dd, 1H, *J* = 3.4, 1.8 Hz, HC = C*H*-CH), 3.64–3.52 (m, 2H, HC = N-C*H*_2_-CH_2_) or 2.53 (ABX signal, 2 × dd, 2H, *J* = 11.5 and 7.1 Hz, HC = N-C*H*a-CH, HC = N-C*H*b-CH), 2.69 (t, 2H, *J* = 7.8 Hz, H_2_N-C*H*_2_-CH_2_) or 2.57 (ABX signal, 2 × dd, 2H, *J* = 12.6 and 6.1 Hz, H_2_N-C*H*a-CH, H_2_N-C*H*b-CH), 2.00–1.19 (m, 7H, 2 × H_labiles_, 2 × CH_2_, CH), 0.95 (d, 3H, *J* = 6.7 Hz, CH_3_). ^13^C NMR (101 MHz, CDCl_3_, ppm) *δ*: 151.5 (Cq_Ar_), 149.7 or 149.5 (CH_imine_), 144.5 (O-*C*H=CH), 113.7 (Cq_Ar_-*C*H = CH), 111.5 (O-CH = *C*H), 68.4 (HC-*C*H_2_-N = CH) or 62.0 (H_2_C-*C*H_2_-N=CH), 48.4 (H_2_N-*C*H_2_-CH) or 42.6 (H_2_N-*C*H_2_-CH_2_), 34.1 (CH), 32.4 and 28.3 (2 × CH_2_), 18.1 (CH_3_). HRMS (*m*/*z*, ES+, [M + H^+^]): C_11_H_18_N_2_O; calculated 195.1492; found 195.1500.

Furfuryldiamine (FDA) (C_11_H_20_N_2_O) (Figure 6). Brown liquid. IR (neat, ν, cm^−1^): 3667–3272 (broad N-H), 2924 and 2851 (C-sp^3^), 1667–1454 (C = C*_aromatic_* and *δ*N-H), 1377, 1336, 1147, 1110, 1075, 1008, 918, 802. ^1^H NMR (400 MHz, CDCl_3_, ppm) *δ*: 7.34 or 7.33 (d, 1H, *J* = 2 Hz, O-CH_ar_), 6.29 or 6.28 (d, 1H, *J* = 3.2 Hz, *H*C = CH-CH), 6.15–6.14 or 6.14–6.13 (m, 1H, HC = C*H*-CH), 3.76 or 3.75 (s, 2H, Cq_ar_-CH_2_), 2.67–2.36 (m, 4H, HN-C*H*_2_-CH_2_ or HN-C*H*_2_-CH, and H_2_N-C*H*_2_-CH_2_ or H_2_N-C*H*_2_-CH), 1.97–1.07 (m, 8H, 3 × H_labiles_, 2 × CH_2_, CH), 0.89 (d, 3H, *J* = 6.8 Hz, CH_3_). ^13^C NMR (101 MHz, CDCl_3_, ppm) *δ*: 154.0 (2 × Cq_Ar_), 141.7 or 141.6 (O-*C*H = CH), 110.0 (Cq_Ar_-*C*H = CH), 106.7 (O-CH = *C*H), 55.5 (HN-*C*H_2_-CH) or 49.4 (HN-*C*H_2_-CH_2_), 48.3 (H_2_N-*C*H_2_-CH) or 42.5 (H_2_N-*C*H_2_-CH_2_), 46.4 or 46.4 (Cq_ar_-*C*H_2_), 36.2 or 33.1 (CH), 32.0 (CH_2_), 27.4 (CH_2_), 18.1 or 17.4 (CH_3_). HRMS (*m*/*z*, ES+, [M + H^+^]): C_11_H_21_N_2_O; calculated 197.1648; found 197.1656.

### 2.4. Amine Hydrogen Equivalent Weight (AHEW or HEW) Calculation

Each experimental HEW was determined by NMR ^1^H titration using benzophenone as the intern reference. To this end, known weights of amine and benzophenone were poured into an NMR tube and 500 µL of deuterated chloroform were added. HEW values were determined according to Equation (4).
(4)$$EEW=\frac{\int PhCOPh*H_{amine}}{\int amine*H_{PhCOPh}}*\frac{m_{amine}}{m_{PhCOPh}}*M_{PhCOPh}$$
where:

∫*_PhCOPh_*:
integration of the benzophenone protons;
∫*_amine_*:
integration of the protons of the amine functions;*H_amine_*:
number of protons of the amine functions;*H_PhCOPh_*:
number of protons of the benzophenone;*m_amine_:*weight of the amine product;*m_PhCOPh_*:
weight of benzophenone;*M_PhCOPh_*:
molecular weight of benzophenone.

### 2.5. Synthesis of Epoxy Thermosets

The amount of hardener for 100 g of epoxy for a theoretical molar ratio of 1:2 between amine and epoxy functions was calculated according to Equations (5) and (6):(5)$$AHEW=\frac{M_{amine}}{n_{fonction\;NH}}$$
(6)$$m_{hardener}=\frac{100\times HEW}{EEW}$$
where AHEW (or HEW) is the amine hydrogen equivalent weight and EEW is the epoxy equivalent weight.

The optimal molar ratio was then determined with the adjustment of Equation (6) by multiplying the hardener mass by various ratio of amine/epoxy. Then, the optimal molar ratio, corresponding to the highest *T_g_*, was determined by recording the *T_g_s* using Differential Scanning Calorimetry (DSC) analysis.

Reactants were then mixed according to the previously determined optimal molar ratio and cured at 80 °C for 2 h to obtain the thermosets.

## 3. Results and Discussion

The aim of this study was to synthesize new pluri-functional and bio-based aminated hardeners containing aromatic moieties. To this end, the reductive amination method was applied to bio-based and non-toxic aromatic aldehydes with a non-toxic diamine (Figure 7). This method brings aromatic moieties, and thus to increase the thermo-mechanical properties of the final material, without having to synthesize aromatic amines which are generally toxic. The pluri-functionality of amine was easily studied by changing the active H number with the selection of the structure of the aldehyde monomer and the reaction conditions.

In this study, we studied the influence of the modification of the active H functionality on the thermo-mechanical properties. The use of dialdehyde monomers increases the active H functionality of the final hardener compared to the initial diamine (respectively six active H versus four), while monoaldehyde leads to three active H. In this view, isophthalaldehyde (IPA) was chosen as the dialdehyde monomer and benzaldehyde as the monoaldehyde reference. All previously cited aldehydes can be produced from biomass and are non-toxic [64,65,66,67,68]. Furfuraldehyde was also chosen, despite its toxicity, in order to compare the final thermoset properties with aromatic ones [69]. Due to its liquid aspect and its presence in the REACH (registration evaluation authorisation and restriction of chemicals) registration list, 2-methylpentane-1,5-diamine (DYTEK^®^A) was chosen as the amine reactant [70]. The presence of branched methyl in its structure decreases the viscosity thanks to the steric hindrance, providing a liquid aspect to the reactant. Moreover DYTEK^®^A could be synthesized by the methylation of natural glutamine [71,72].

### 3.1. Hardeners Synthesis

Each aldehyde reactant was respectively added dropwise, in an excess of initial amine reactant, in order to avoid the oligomer formation. To simplify the procedure, we chose NaBH_4_ as a reducing agent for this study. However, this reducing step can be easily performed with an industrial process such as catalytic hydrogenation [31]. Only IPA-based amine was also synthesized in one-pot conditions, by mixing entirely aldehyde at the same time as amine, to favor the dimerization (i.e., in order to obtain a higher functionality). The aims of this other one-pot method was to study the influence of the increase of the active hydrogen functionality to six on the network properties. All amine characterizations are summarized in Table 1. 

The reductive amination method obtains full conversion of aldehyde during the imine synthesis step and then, full conversion of imine during the second step (spectra given in Appendix A). All amine hardeners from IPA, benzaldehyde and furfural were successfully synthesized.

The imines synthesis and their reduction may be easily monitored by the appearance and disappearance of the imine signal from both FTIR and ^1^H NMR spectra (Figure 8). For instance, in the case of IPA-based hardener synthesis (IPTA), the reduction of imine moieties to amine functions may be observed with the disappearance of C = N stretching band at 1643 cm^−1^. The ^1^H NMR spectrum changes considerably after the reduction step. The disappearance of the –*CH* signal corresponding to the imine proton at 8.25 ppm is observed and then confirmed by the appearance of a singlet signal at 3.77 ppm, corresponding to –*CH_2_* signal of the reduced imine function. Moreover, the disappearance of the –*CH_2_* signal corresponding to the protons in α position of C = N bond designated as 5 and 5′ is shown in Figure 8. The reduction step induces the absence of conjugated system involving the imine bond. Consequently, the aromatic signals of the formed amines are shifted from the area 8.00–7.36 to the area 7.27–7.18 ppm.

^1^H NMR spectra of the three synthesized amines from IPA (named IPTA1), benzaldehyde (named BDA) and furfural (named FDA) are displayed in Figure 9. The IPA-based hardener synthesized in one-pot conditions (IPTA2) shows a similar spectrum to IPTA1, with different integration values (spectra given in Appendix A). The signal corresponding to the Cq_Ar_-C*H*_2_ protons in α position of secondary amine appears at 3.75 ppm with a singlet (designated as 1 in Figure 9). Then, the signals of the other α-CH_2_ of the secondary and primary amine moieties are overlapped at 2.51 ppm (designated as 2 in Figure 9). Moreover, the signals of the amine from the A addition are different from the signals of the amine from the B addition but are found in the same overlapped signal. Thus, there are at least four signals overlapped at 2.51 ppm. Due to this overlap, the A and B addition ratio could not have been determined thanks to the ^1^H NMR spectra of the final amine monomers. However, before the reduction step, the α-CH_2_ of the imine signal shifted to 3.50 ppm due to the conjugated system involving the imine bond. This signal is split into two different signals for A and B additions, determining the proportion of each other (spectra given in Appendix A).

The characterizations of the new bio-based amine hardeners named IPTA1, IPTA2, BDA and FDA are summarized in Table 1 (DSC and TGA thermograms are given in Appendix A). A and B additions were obtained in similar proportions, with an average ratio of 50:50 (determined using the ^1^H NMR spectrum of imine intermediates). The experimental HEW was determined by NMR titration, as reported in the Materials and Methods section. Experimental and theoretical results were almost similar. The higher glass transition temperature (*T_g_*) of IPTA2 compared to IPTA1 confirms the presence of more dimers in the IPTA2 hardener than in the IPTA1. The viscosity value follows the same trend with a higher value for IPTA2 than IPTA1. Due to the higher content of aromatic moieties in dimeric structure and so in IPTA2 hardener, *T_g_* and *T_d_5%* of IPTA2 are higher than IPTA1 monomeric structure. BDA and FDA both exhibit similar *T_g_* and *T_d_5%* values, showing similar behaviors for their thermomechanical properties. IPTA1, BDA and FDA exhibit a liquid aspect with low viscosities lower than 0.6 Pa s at 22 °C close to the water aspect, which are interesting properties for epoxy-amine formulations. The comparison of IPTA1 and IPTA2 hardeners shows an increase of viscosity with the number of dimers.

### 3.2. Thermoset Syntheses

Synthesized amines were then used to synthesize epoxy thermosets (also named P-materials) with different epoxy monomers. Hence, bulk materials (parallelepiped shape) were obtained by the curing of the synthesized amines with epoxy monomers, using previously-determined optimal ratios (method described in the Experimental section, DSC thermograms and optimal ratio given in Appendix A). First, hardeners were reacted with diglycidyl ether of bisphenol a (DGEBA), as a petro-sourced epoxy reference. These thermosets can be compared to literature results describing the networks’ characteristics of MXDA- and DYTEK^®^A-based materials from DGEBA as an epoxy part [73,74,75]. *m*-Xylylenediamine (MXDA) is a petro-sourced arylamine hardener currently used in the industrial field of epoxy coatings due to its high reactivity [76]. It is interesting to compare MXDA and IPTA structures due to their similar dibenzyl center. We could observe the influence of the aliphatic chain addition provided by DYTEK^®^A using DYTEK^®^A-DGEBA thermoset results as data references. In the same way, thermosets (also named bio-materials) were then synthesized using the diglycidyl ether of vanillin alcohol (DGEVA) [77] as a bio-based epoxy derived from vanillin in order to increase the bio-based carbon content. Then, thermo-mechanical properties and chemical resistance in THF of each optimal bulk network are determined and summarized in Table 2.

Epoxy and amine reactants were mixed and then cured at 80 °C for 2 h, with reactant amounts corresponding to the respective optimal ratios. The end of the crosslinking reaction was confirmed by DSC analyses, with no residual enthalpy signal on each thermogram. Furthermore, high-gel content values (>90%), corresponding to a highly cross-linked material, confirmed the full conversion for each thermoset.

The thermal stabilities were determined using TGA under nitrogen steam. The 5% weight loss (*T_d_5%*) temperature and the char yields at 600 °C were recorded (Figure 10). By comparing IPTA1-based and IPTA2-based materials, results followed the same trend exhibiting similar *T_d_5%* and char yield values. Furthermore, the P-materials showed higher thermal resistance with slightly higher *T_d_5%* values around 350 °C against 315 °C for bio-materials, keeping however good thermal resistance. On the contrary, a higher char yield at 600 °C was observed for bio-based materials, meaning higher thermal resistance for high temperatures. This can be explained by the absence of the geminal dimethyl bridge on the bio-based epoxy structure, which has a low thermal stability. TGA results concluded that the slightly molecular weight difference between IPTA1 and IPTA2 has no impact on thermal stability. IPTA-based materials showed slightly higher char yields than BDA-based material with a residual mass of 3%–4% higher. FDA-based materials exhibited the highest char yield, meaning that furan moieties showed higher thermal resistance than benzyl moieties. All P-materials showed a higher thermal stability than MXDA-DGEBA material (MXDA-ref).

The glass transition temperature values (*T_g_*) were recorded by DSC and then compared to the alpha transition temperature values (*T_α_*), which were determined by DMA analyses, corresponding to the mechanical manifestation of *T_g_* (DMA in Figure 11, DSC thermograms given in Appendix A). The transition from vitreous state to rubbery state induces a module loss, and thus the maximum value of the tan *δ* curve as a function of the temperature corresponds to the *T_α_*. Results showed that *T_g_*s and *T_α_*s followed the same trend for each thermoset with each of narrowed tan *δ* peaks, suggesting that the materials are homogeneous. Overall, fully bio-based thermosets exhibited a lower *T_g_* value than the P-material references, with a difference of 30 to 40 °C. This decrease of *T_g_* value is due to the presence of methylene and methoxy moieties in DGEVA structure, which behaved as spacers and repulse polymeric chains for each other, thereby providing flexibility and thus lower *T_g_*.

DSC results showed a difference of 10 °C between IPTA1- and IPTA2-based thermosets in favor to IPTA1-based ones due to the reduced molecule weight for the thermosets from IPTA1, that displayed shorter backbones than IPTA2. Then, the IPTA-based networks were compared to BD-P reference. BD-P is based from BDA which is a hardener with only three active hydrogen functions (one secondary and one primary amine) while IPTA-based hardeners show similar backbone structure containing at least two alkyl chains with six active hydrogens (two secondary and two primary amines). Due to the presence of more alkyl chains containing –N*H* functions in IPTA structure, the aromatic moieties were directly incorporated in the polymeric chain compared to the BDA structure, which induced alkyl polymeric chains with dandling aromatic moieties (Figure 12). Moreover, the six –N*H* reactive functions of IPTA increased the cross-linking density (*ν*′) values compared to BDA (respectively 1311 and 460 mol⋅m^−3^ for P-materials, and 997 and 659 mol⋅m^−3^ for bio-materials). Then, BDA-based and FDA-based thermosets exhibited similar *T_g_* and *T_α_*, without any distinction between furan and benzyl moieties. However, the cross-linking density values showed that thermosets from furan have more compact networks than benzyl-based thermosets (with respectively 460 and 123 mol⋅m^−3^ for P-materials, and 407 and 91 mol⋅m^−3^ for bio-materials). Moreover, those results are confirmed by the swelling index values, which are inversely proportional to the cross-linking density. A compact network involves a lower solvent ingress and thus a lower swelling index.

Overall, the comparison of *T_g_* and *T_α_* for these four networks concluded that the higher the amount of active hydrogens, the higher *T_g_* and *T_α_*. However, by comparing IPT1-P, IPT2-P and BD-P with DYTEK^®^A-ref and MXDA-ref, the decrease of *T_g_* and *T_α_* could be noticed. In the case of BD-P, the addition of benzyl moiety induced a loss of one active hydrogen in the structure, reducing the cross-linking density and thus increasing the flexibility of the network (*ν*′_BDA_ = 460 mol⋅m^−3^ against 1146 for MXDA-ref). For IPTA-based thermosets, the active hydrogen functionality up to six, increased the cross linking-density (*ν*′_IPTA1_ = *ν*′_IPTA2_ = 1311 mol⋅m^−3^). However, the presence of aliphatic chains induced a higher molecular length, allowing higher microscopic deformation and exhibiting lower *T_g_* (in increasing order of molecular length: *T_g_* of MXDA-ref = 116 °C ≈ *T_g_* of DYTEK^®^A-ref, *T_g_* of IPT1-P = 99 °C, and *T_g_* of IPT2-P = 89 °C).

Finally, the storage modulus of vitreous (E’_glassy_) and rubbery (E’_rubbery_) domains were determined respectively at *T*_(*α*−80)_ and *T*_(*α*+80)_ using DMA analyses, allowing to obtain macroscopic deformation information. The storage modulus (E’_rubbery_) is also linked to the cross-linking density, according to the rubber-elasticity theory [78]. The results showed similar storage modulus in the elastic domain for IPTA-based thermosets with an E’_rubbery_ order of magnitude of 10^7^ Pa, which corresponds to a high mechanical strength for thermosets compared to classical high performance thermosets such as MXDA-ref and DYTEK^®^A-ref materials [73,74,75]. It is interesting to note that similar mechanical strength was obtained with a lower *T_g_* in the case of IPTA-based materials. These results were induced by the particular structure of IPTA compared to MXDA and DYTEK^®^A references. IPTA exhibits indeed six –N*H* active functions, increasing the cross-linking density and thus the mechanical strength at the macroscopic scale, while the longer molecular length, induced by the presence of two alkyl chains in IPTA backbone, decreases the *T_g_* at the microscopic scale. In the case of BD-P and FD-P, which exhibits three –N*H* active functions, results showed a lower order of magnitude of 10^6^ Pa with a lower value for FD-P (E’_rubbery_ of BD-P = 4.84 × 10^6^ Pa and E’_rubbery_ of FD-P = 1.30 × 10^6^ Pa), meaning a lower mechanical strength than IPT1-P, IPT2-P and BD-P. The slightly lower E’_rubbery_ value could be assumed by the higher aromaticity of benzene moieties than furan ring [79] which induces higher π-stacking in BDA-based materials [80,81]. By comparing BD-P and FD-P to DYTEK^®^A-ref, it should be noted that the loss of one –N*H* active function induced a lower mechanical strength due to the decrease of cross-linking density. On the contrary, IPTA-based materials exhibited similar storage modulus thanks to higher cross-linking density despite the lower *T_g_* value.

## 4. Conclusions

New pluri-functional amine hardeners based on various bio-based aldehydes were synthesized using the reductive amination method. Three amine monomers were obtained: IPTA, BDA and FDA from isophthalaldehyde, benzaldehyde and furfuraldehyde, respectively. IPTA exhibits six active amine hydrogens versus three for BDA and FDA. The molecular weight of IPTA and thus its properties could be modified by changing reaction conditions (dropwise addition or one-pot conditions). All hardeners exhibited no or slight color, without odor. As expected, theses hardeners are liquid with low viscosity (IPTA1, BDA and FDA < 0.6 Pa s at 22 °C), a lower viscosity than our previously synthetized β-hydroxylamine hardeners (>300 Pa s at 50 °C) [21]. Due to their low viscosities, the epoxy-amine mixtures were easily homogenized and rapidly cured with an optimal epoxy-amine ratio, using DGEBA as petro-sourced epoxy reference and DGEVA as bio-based epoxy monomer. Synthesized thermosets showed great thermo-mechanical properties with higher results for IPTA-based materials, which showed similar mechanical strength and cross-linking density than MXDA-ref and DYTEK^®^A-ref (*ν*′_IPTA1_ = *ν*′_IPTA2_ = 1 311 mol⋅m^−3^ against 1 146 for MXDA--ref and E’_rubbery_ of IPTA1 = E’_rubbery_ of IPTA2 = 1.38.10^7^ Pa, E’_rubbery_ of DYTEK^®^A-ref = 2.90.10^7^ Pa, E’_rubbery_ of MXDA-ref = 1.80.10^7^ Pa). However, a lower *T_g_* around 95 °C was observed for IPTA-based materials (against *T_g_* = 116 °C for MXDA-ref and *T_g_* = 115 °C for DYTEK^®^A-ref).

In fact, the functionality of six active hydrogens of IPTA led to unique, highly cross-linked systems exhibiting lower *T_g_* due to the presence of two alkyl chains in the IPTA backbone, allowing microscopic scale deformation, and keeping high mechanical strength at the macroscopic scale, similar to industrial references.

## Figures and Tables

**Figure 1 molecules-24-03285-f001:**
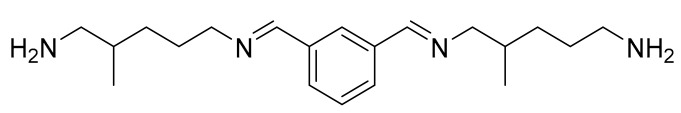
Isophtalatediiminediamine (IPDIDA).

**Figure 2 molecules-24-03285-f002:**
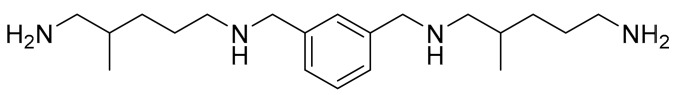
Isophtalatetetramine (IPTA).

**Figure 3 molecules-24-03285-f003:**
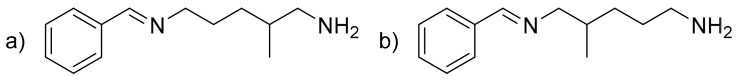
(**a**) Benzylimineamine (BIA) from A addition and (**b**) BIA from B addition.

**Figure 4 molecules-24-03285-f004:**
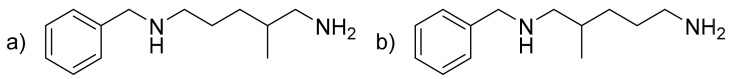
(**a**) Benzyldiamine (BDA) from A addition and (**b**) BDA from B addition.

**Figure 5 molecules-24-03285-f005:**
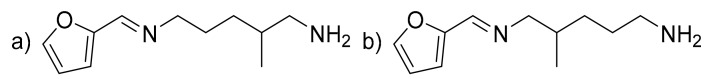
(**a**) Furfurylimineamine (FIA) from A addition and (**b**) FIA from B addition.

**Figure 6 molecules-24-03285-f006:**
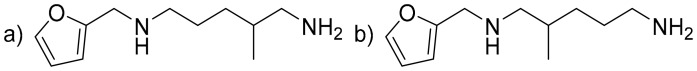
(**a**) Furfuryldiamine (FDA) from A addition and (**b**) FDA from B addition.

**Figure 7 molecules-24-03285-f007:**
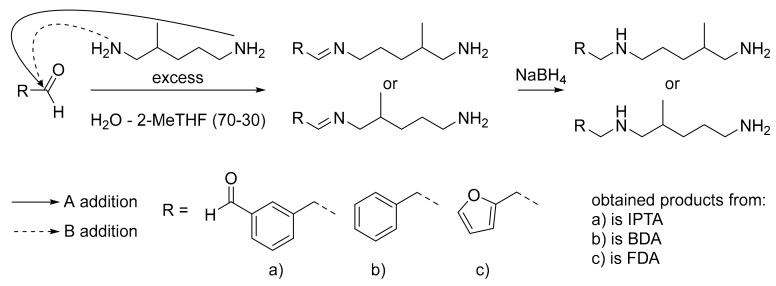
Amines syntheses from DYTEK^®^A as amine reactant and (**a**) isophthalaldehyde (IPA), (**b**) benzaldehyde and (**c**) furfural, as aldehyde monomers.

**Figure 8 molecules-24-03285-f008:**
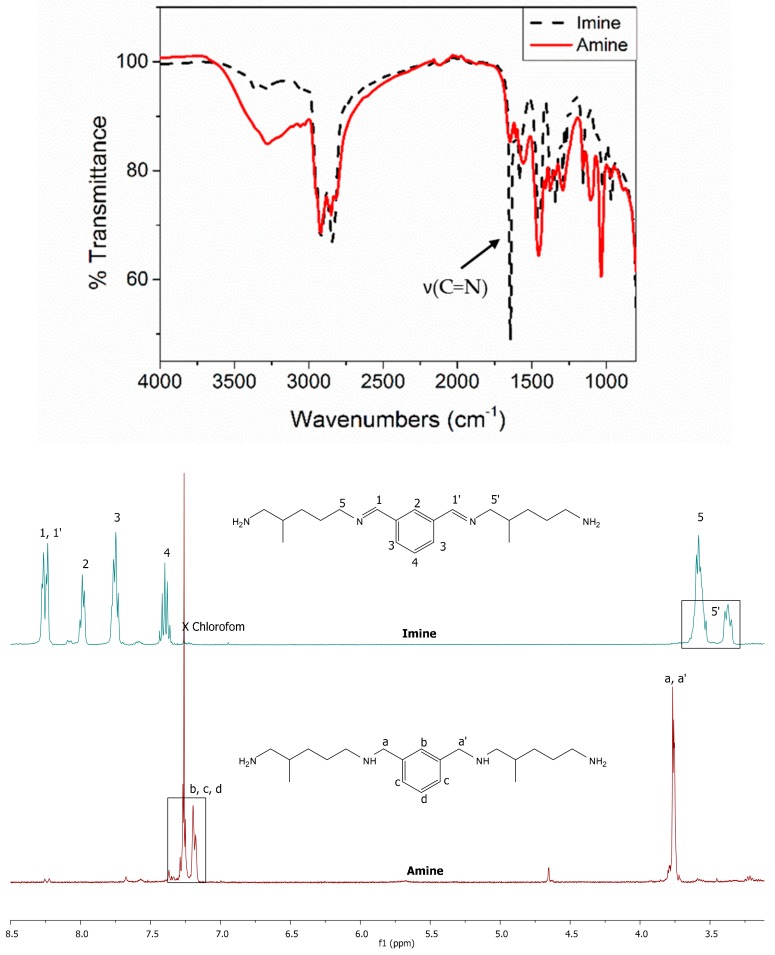
Imine signal disappearance of IPDIDA, the corresponding imine of IPTA amine monomer, followed by FTIR (up), and ^1^H NMR in CDCl_3_ (down).

**Figure 9 molecules-24-03285-f009:**
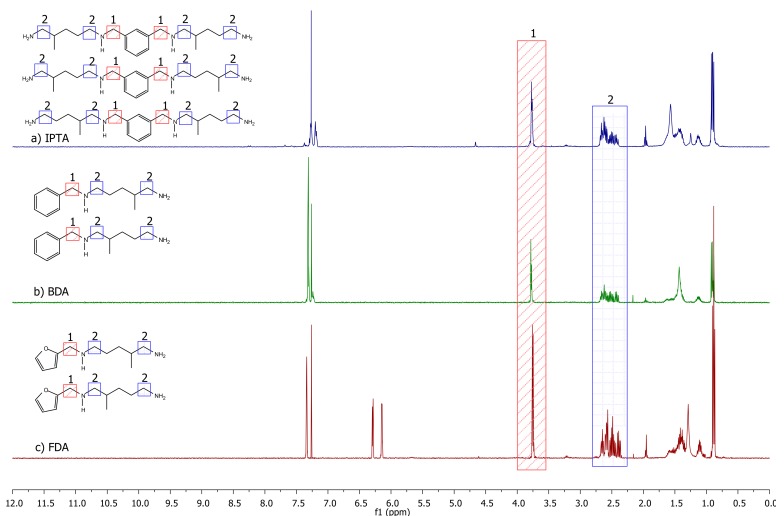
^1^H NMR spectra of (**a**) IPTA2, (**b**) BDA and (**c**) FDA in CDCl_3_.

**Figure 10 molecules-24-03285-f010:**
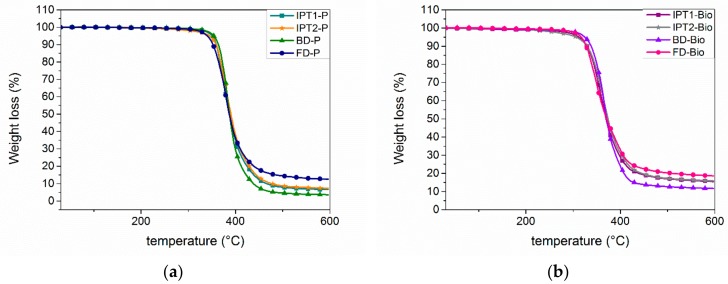
**Thermogravimetry Analysis** (TGA) measurements of the thermosets from new hardeners and (**a**) DGEBA or (**b**) DGEVA as epoxy.

**Figure 11 molecules-24-03285-f011:**
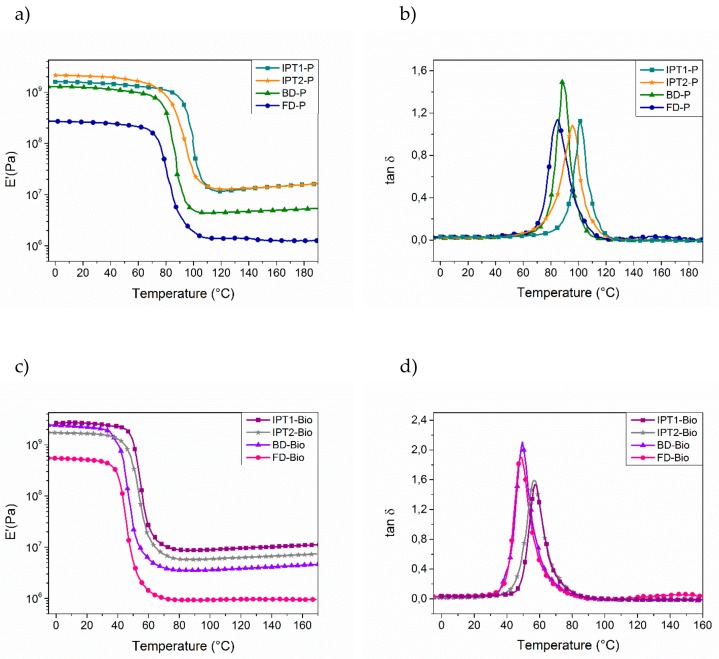
**Dynamic Mechanical Analysis** (DMA) thermograms (under air, 1 Hz, 3 K⋅min^−1^): (**a**) *E*′ curves of materials cured with DGEBA as epoxy; (**b**) Tan *δ* curves of materials cured with DGEBA as epoxy; (**c**) *E*′ materials with DGEVA as epoxy; (**d**) Tan *δ* curves of materials cured with DGEVA as epoxy. All thermosets were cured with respective optimal ratios.

**Figure 12 molecules-24-03285-f012:**
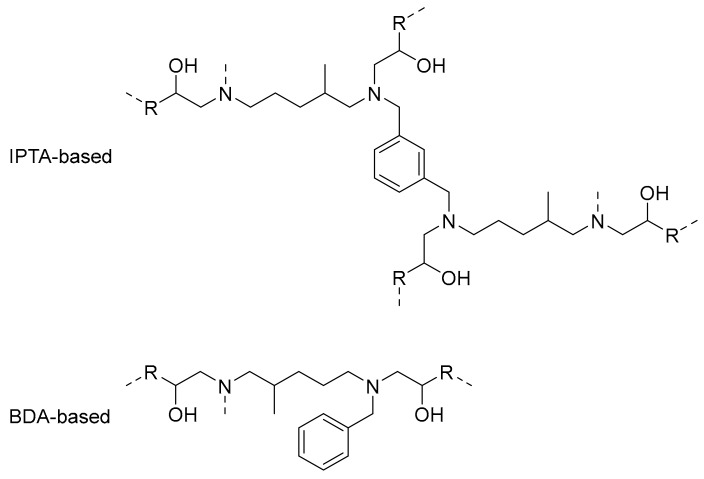
Theoretical structures of polymeric chain units for IPTA-based and BDA-based material.

**Table 1 molecules-24-03285-t001:** Amine characterizations.

Amine Nomenclatures	A: B Addition Ratio ^a^	HEW_th_ (g⋅eq^−1^)	HEW_exp_^b^ (g⋅eq^−1^)	Amine Viscosity (Pa⋅s at 22 °C)	*T_d_5%* (°C)	Char Yield^c^ (%)	*T_g_*^d^ (°C)
IPTA1	51: 49	56	57	0.57	173	1	− 57
IPTA2	45: 55	56	59	1.40	204	4	− 35
BDA	48: 52	69	67	0.03	115	3	− 81
FDA	48: 52	65	66	0.02	126	2	− 84

^**a**^ Determined at the end of imine synthesis reaction; ^**b**^ determined by NMR ^1^H titration; ^**c**^ at 600 °C; ^**d**^ mid values. HEW: hydrogen equivalent weight.

**Table 2 molecules-24-03285-t002:** Thermoset characterizations.

Thermosets	Composition Amine-Epoxy	*T_d_5%* (°C)	Char Yield ^c^ (%)	*T_g_*^d^ (°C)	*T_α_* (°C)	*ν*′ (mol⋅m^−3^)	E’_glassy_ (Pa)	E’_rubbery_ (Pa)	SI^e^ (%)	GC (%)
DYTEK^®^A-ref ^a^	DYTEK^®^A-DGEBA	-	-	115	127	-	-	2.90.10^7^	-	-
MXDA-ref ^b^	MXDA-DGEBA	333	7	116	130	1 146	1.2.10^8^	1.80.10^7^	89	99
IPT1-P	IPTA1-DGEBA	351	7	99	101	1 311	1.6.10^9^	1.38.10^7^	32	100
IPT2-P	IPTA2-DGEBA	345	7	89	95	1 311	2.14.10^9^	1.38.10^7^	93	100
BD-P	BDA-DGEBA	354	4	77	88	460	1.29.10^9^	4.84.10^6^	123	97
FD-P	FDA-DGEBA	340	13	76	85	123	2.71.10^8^	1.30.10^6^	122	99
IPT1-Bio	IPTA1-DGEVA	312	16	59	57	997	2.7.10^9^	1.05.10^7^	41	100
IPT2-Bio	IPTA2-DGEVA	307	16	57	57	657	1.72.10^9^	6.92.10^6^	108	96
BD-Bio	BDA-DGEVA	325	12	45	50	407	2.41.10^9^	4.29.10^6^	134	95
FD-Bio	FDA-DGEVA	317	19	46	49	91	5.45. 10^8^	9.59.10^5^	125	100

ref ^**a**^ literature results from Fu. et al. [78] and Meis et al. [79]; ref ^**b**^ literature results from Faye. et al. [77]; ^**c**^ at 600 °C; ^**d**^ mid values; ^**e**^ in THF.

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
