# Peer review of "Synthesis of Pluri-Functional Amine Hardeners from Bio-Based Aromatic Aldehydes for Epoxy Amine Thermosets"

_molecules, 2019, doi:10.3390/molecules24183285_

Round 1
Reviewer 1 Report
The authors describe the design of amine hardeners for epoxy resins that are fully bio-based and because of low viscosity shall enable effective mixing and hence crosslinking.
There are a couple of points to address to improve the quality of the manuscript. These are in order of occurrence in the manuscript and not in order of importance.
Abstract: The abstract contains a large number of abbreviations that cannot be understood without having read the article. The abstract should be understandable as stand-alone, and should be revised accordingly. Last sentence of abstract: high rigidity and low Tg are typically opposing concepts. Please give a short explanation here how that is possible, and state so as well in the discussion + conclusion parts Add all used abbreviations to the abbreviations section as well. Experimental part: Give information on state and city for the companies providing chemicals and equipment. Clearly state how you define Tα - which transition do you refer to, how is it determined (-> experimental part), why is it important. The term Tα is typically only used in very specialized papers, and different authors use different nomenclature/definitions. Whenever that is the case, (i.e. a non widely accepted concept is used) the authors should take extra care in explaining the concept and significance of the results. Format: all over the manuscript, a dot (such as °C.min...) is wrongly used, please correct to a central dot. (°C⋅min) Swelling Indices: Swelling is defined as change of volume, here a change of mass (i.e. solvent uptake) is reported. State the volume changes (as these according to theory can be discussed further for crosslink density, which plays a role here, and should be incoroprated in the revised manuscript). NMR assignments: Do not state coupling constants for aromatic protons, as these are no constants (try measure them on a spectrometer with a different field strength!!). Don't assign e.g. "triplett" or the like either for the aromatics, as again these depend on the spectrometer used. Acceptable are either "multiplett" or "apparent [dublet/triplett]" etc. line 315ff: Why does water solubility prevent a purification???? There are many ways to purify compounds, including chromatography etc. It would be good to show some of the mass specs, e.g. in the supporting information, of the new hardeners line 407 ff: Sentence "...exhibit a Tg value twice more less..." does not make sense. Even when comparing e.g. 100 °C to 40 °C, 40 °C would not be less than half of 100 °C (-> ever heard of absolute temperatures?). Talk about absolute differences (as in the subsequent sentence) Figures 10 +12: The legend as well as the x-axis have been cut and are not fully readable Please check the language - it is altogether OK, but some language editing is required, e.g. "exhibits liquid aspect" (?? are liquid??). PLease crosscheck the full manuscript. Conclusion: too many generalities / imprecise wordings, e.g. i) "really lower" (?), ii) "good thermo-mechanical properties with better..." [don't use such word as "good" or "better" -> this refers to fitting to a specific requirement, e.g. a lower Young's modulus could be better fitting to a certain application. State numbers and or specify in terms of "higher than..."], iii) line 474: 115 for references??? Please re-write.Author Response
Reviewer 1
The authors describe the design of amine hardeners for epoxy resins that are fully bio-based and because of low viscosity shall enable effective mixing and hence crosslinking. There are a couple of points to address to improve the quality of the manuscript. These are in order of occurrence in the manuscript and not in order of importance.
The abstract contains a large number of abbreviations that cannot be understood without having read the article. The abstract should be understandable as stand-alone, and should be revised accordingly. We fully agree with the Reviewer #1 and we have made changes in the paper accordingly in the abstract :
“Then, these pluri-functional amines were added to petro-sourced (DGEBA) or bio-based (DGEVA) epoxy monomers to form thermosets by step growth polymerization. Due to their low viscosity, the epoxy-amine mixtures were easily homogenized and cured more rapidly compared to the use of more viscous hardeners (<0.6 Pa⋅s at 22 °C). After curing, the thermo-mechanical properties of the epoxy thermosets were determined and compared. The isophtalalatetetramine (IPTA) hardener, with a higher number of amine active H, led to thermosets with higher thermo-mechanical properties (glass transition temperatures (Tg and Tα) were around 95 °C for DGEBA-based thermosets against 60 °C for DGEVA-based thermosets) than materials from benzyldiamine (BDA) or furfuryldiamine (FDA) that contain less active Hydrogens (Tg and Tα around 77 °C for DGEBA-based thermosets and Tg and Tα around 45 °C for DGEVA-based thermosets).”
Last sentence of abstract: high rigidity and low Tg are typically opposing concepts. Please give a short explanation here how that is possible, and state so as well in the discussion + conclusion parts. We understand the concern raised by Reviewer. Indeed, our explanations were not clear enough and we developed them in order to improve understandability and avoid any confusion. Indeed, the low Tg is based on nanoscale mobility of chains, whereas modulus (we removed the term rigidity to avoid any confusion) depends also strongly on macroscale properties that are modified by cross-linking density. Me modified manuscript accordingly:
In the abstract:
By comparing to industrial hardener references, IPTA possesses 6 active hydrogens allowing to obtain high cross-linked systems, similar to industrial references, and longer molecular length due to the presence of two alkyl chain, leading respectively to high mechanical strength with lower Tg.
In the discussion:
“However, the presence of aliphatic chains induced a higher molecular length, allowing higher microscopic deformation and exhibiting lower Tg (in increasing order of molecular length: Tg of MXDA-ref = 116 °C ≈ Tg of DYTEK®A-ref, Tg of IPT1-P = 99 °C, and Tg of IPT2-P = 89 °C).”
And: “The results showed similar storage modulus in the elastic domain for IPTA-based thermosets with an E’rubbery order of magnitude of 107 Pa, which corresponds to a high mechanical strength for thermosets compared to classical high performance thermosets such as MXDA-ref and DYTEK®A-ref materials.[77-79] It is interesting to note that similar mechanical strength were obtained with a lower Tg in the case of IPTA‑based materials. These results were induced by the particular structure of IPTA compared to MXDA and DYTEK®A references. IPTA exhibits indeed 6 –NH active functions, allowing to increase the cross‑linking density and thus the mechanical strength at the macroscopic scale, while the longer molecular length, induced by the presence of two alkyl chains in IPTA backbone, allows to decrease the Tg at the microscopic scale. In the case of BD-P and FD-P, which exhibit 3 –NH active functions, results showed a lower order of magnitude of 106 Pa with a lower value for FD-P”
In the conclusion:
“In fact, the 6 active hydrogen functionality of IPTA allowed to obtain unique highly cross‑linked systems exhibiting lower Tg due to the presence of two alkyl chain in the IPTA backbone, allowing microscopic scale deformation, and keeping high mechanical strength at the macroscopic scale, similar to industrial references.”
Add all used abbreviations to the abbreviations section as well. We have added all used abbreviations accordingly
Abbreviations: AHEW or HEW, amine hydrogen equivalent weight, BDA, benzyldiamine, BIA, benzylimineamine, DGEBA, diglycidyl ether of bisphenol A, DGEVA, diglycidyl ether of vanillyl alcohol, DMA, dynamic mechanical analysis, DSC, differential scanning calorimetry, EEW, epoxy equivalent weight, FDA, furfuryldiamine, FIA, furfurylimineamine, FTIR or IR, Fourier transform infrared, GC, gel content, IPA, isophtalaldehyde, IPDIDA, isophtalalatediiminediamine, IPTA, isophtalalatetetramine, MXDA, m‑xylylenediamine, NMR, nuclear magnetic resonance, PACM, 4,4’‑methylenebis(cyclohexylamine), ppm, parts per million, SI, swelling index, THF, tetrahydrofuran, TGA, thermogravimetric analysis.
Experimental part: Give information on state and city for the companies providing chemicals and equipment. Information on state and city of supplier companies have been given
“Benzaldehyde (purity ³99.0%), 1,5-diamino-2-methylpentane as DYTEK®A (purity ³99.0%), furfural (purity ³99.0%), 1,6-hexanediamine (purity ³98.0%), terephthaldehyde (purity ³99.0%) were purchased from Sigma-Aldrich (St. Quentin Fallavier, France). 2‑methyltetrahydrofuran (purity ³99.0%), was purchased from Alfa Aesar (Kandel, Germany). Isophthalaldehyde (purity ³98.0%) was purchased from TCI (Zwijndrecht, Belgium). Ethyl acetate (purity ³99.9%) was purchased from VWR Chemicals (Fontenay-sous-Bois, France). Deuterated solvents were obtained from Sigma Aldrich for NMR study.”
“1H and 13C NMR analyses were recorded on a 400 MHz Bruker Aspect NMR spectrometer at 23 °C (Rheinstetten, Germany), in deuterated solvents.”
“Viscosities measurements were performed at 22 °C on the AR-1000 rheometer (TA Instruments, New Castle, DE, USA).”
“Fourier transform infrared (FTIR) spectra were recorded using a Thermo Scientific Nicolet 6700 FTIR spectrometer with “diamond ATR” equipment (Waltham, MA, USA) in transmittance and with a band accuracy of 4 cm−1.”
“Thermogravimetric Analyses (TGA) were recorded using a Netzsch F1-Libra analyzer (Selb, Germany) at a heating rate of 20 °C.min−1 from 25 to 600 °C (nitrogen stream).”
“Differential scanning calorimetry (DSC) measurements were performed with the use of a Netzsch DSC200F3 calorimeter F3 (Selb, Germany, indium calibration, nitrogen stream).”
“Dynamic Mechanical Analyses (DMA) were performed on Metravib DMA 25 with Dynatest 6.8 software (TA Instruments, New Castle, DE, USA). “
Clearly state how you define Tα - which transition do you refer to, how is it determined (-> experimental part), why is it important. The term Tα is typically only used in very specialized papers, and different authors use different nomenclature/definitions. Whenever that is the case, (i.e. a non widely accepted concept is used) the authors should take extra care in explaining the concept and significance of the results. We understand the issue raised by the Reviewer. We have modified the manuscript as follows:
“The glass transition temperature values (Tg) were recorded by DSC and then compared to the alpha transition temperature values (Tα), which were determined by DMA analyses, corresponding to the mechanical manifestation of Tg (DMA in Figure 12, DSC thermograms given in supporting information, part 8). The transition from vitreous state to rubbery state induces a module loss, and thus the maximum value of the tan δ curve as a function of the temperature corresponds to the Tα.”
Format: all over the manuscript, a dot (such as °C.min...) is wrongly used, please correct to a central dot. (°C⋅min) We have modified the manuscript accordingly.
Swelling Indices: Swelling is defined as change of volume, here a change of mass (i.e. solvent uptake) is reported. State the volume changes (as these according to theory can be discussed further for crosslink density, which plays a role here, and should be incoroprated in the revised manuscript). We understand the point raised by the Reviewer. In fact the objective is to access to gel content by solvent solubilization of soluble chains. During this characterizations we also measure solvent uptake, which allow also to measure swelling. Hence, in order to favor the solubilization of free species into the solvent to determine the gel content, a large amount of THF (25 mL) was added to the flask containing only 25 mg of material. Thus, the volume increase induced by the swelling of the thermosets sample is difficult to measure efficiently. That’s why the calculation method using the mass of the thermoset was here used. This method is used in the thermoset field literature (cited following), allowing to obtain a correct evaluation of the swelling index. -Q. Yang et al., Biodegradable cross-linked poly(trimethylene carbonate) networks for implant applications: Synthesis and properties, Polymer, 2013, 54, 2668-2675. Kasetaite et al., J. Effect of Selected Thiols on Cross-Linking of Acrylated Epoxidized Soybean Oil and Properties of Resulting Polymers, Polymers 2018, 10, 439.
We have modified the manuscript accordingly
“Swelling indices (SI) were measured with sample of approximately 25 mg, which was placed in 25 mL of tetrahydrofuran (THF) for 24 h.”
NMR assignments: Do not state coupling constants for aromatic protons, as these are no constants (try measure them on a spectrometer with a different field strength!!). Don't assign e.g. "triplett" or the like either for the aromatics, as again these depend on the spectrometer used. Acceptable are either "multiplett" or "apparent [dublet/triplett]" etc. We agree with Reviewer and we have modified manuscript accordingly.
“1H NMR (400 MHz, CDCl3, ppm) δ: 8.26 (s, 1H, CHimine), 8.23 (s, 1H, CHimine), 7.99 (m, 1H, HAr), 7.75 (m, 2H, 2 x HAr), 7.40 (m, 1H, HAr),”
line 315ff: Why does water solubility prevent a purification???? There are many ways to purify compounds, including chromatography etc. We understand this point. All references to vanillin and 3,4-dihydroxybenzaldehyde have been removed from this manuscript. However, due to the high polarity of amine harderners containing hydroxyl moieties, no purification is possible by chromatography. We have also tried deactivated column without obtaining effective purification.
It would be good to show some of the mass specs, e.g. in the supporting information, of the new hardeners We have modified manuscript accordingly.
line 407 ff: Sentence "...exhibit a Tg value twice more less..." does not make sense. Even when comparing e.g. 100 °C to 40 °C, 40 °C would not be less than half of 100 °C (-> ever heard of absolute temperatures?). Talk about absolute differences (as in the subsequent sentence) We agree with Reviewer and have modified manuscript accordingly:
“Overall, fully bio-based thermosets exhibited a lower Tg value than the P‑material references, with a difference of 30 to 40 °C.”
Figures 10 +12: The legend as well as the x-axis have been cut and are not fully readable We have modified manuscript accordingly.
Please check the language - it is altogether OK, but some language editing is required, e.g. "exhibits liquid aspect" (?? are liquid??). PLease crosscheck the full manuscript. We have proof read and thoroughly corrected the manuscript.
Conclusion: too many generalities / imprecise wordings, e.g. i) "really lower" (?), ii) "good thermo-mechanical properties with better..." [don't use such word as "good" or "better" -> this refers to fitting to a specific requirement, e.g. a lower Young's modulus could be better fitting to a certain application. State numbers and or specify in terms of "higher than..."], iii) We thank Reviewer 1 for his comment and we have improved conclusion following his propositions:
“The molecular weight of IPTA and thus its properties could be modified by changing reaction conditions (dropwise addition or one-pot conditions). All hardeners exhibited no or slight color, without odor. As expected, theses hardeners are liquid with low viscosity (IPTA1, BDA and FDA <0.6 Pa⋅s at 22 °C), lower viscosity than our previously synthetized β‑hydroxylamine hardeners (>300 Pa⋅s at 50 °C).[21] Due to their low viscosities, the epoxy‑amine mixtures were easily homogenized and rapidly cured with an optimal epoxy‑amine ratio, using DGEBA as petro-sourced epoxy reference and DGEVA as bio‑based epoxy monomer. Synthesized thermosets showed great thermo-mechanical properties with higher results for IPTA-based materials, which showed similar mechanical strength and cross‑linking density than MXDA‑ref and DYTEK®A-ref (ν’IPTA1 = ν’IPTA2 = 1 311 mol⋅m−3 against 1 146 for MXDA‑‑ref and E’rubbery of IPTA1 = E’rubbery of IPTA2 = 1.38.107 Pa, E’rubbery of DYTEK®A‑ref = 2.90.107 Pa, E’rubbery of MXDA‑ref = 1.80.107 Pa). However, a lower Tg around 95 °C was observed for IPTA-based materials (against Tg = 116 °C for MXDA-ref and Tg = 115 °C for DYTEK®A‑ref)”
line 474: 115 for references??? Please re-write. We have rewritten the text accordingly.
“(Tg MXDA-ref= 116 °C and Tg DYTEK®A-ref =115 °C).”
Reviewer 2 Report
The manuscript submitted by Mora and co-workers describes the synthesis of new bio-based pluri-functional amine hardeners for the synthesis of epoxy thermosets. The authors clearly presented the objectives of the work and the global concerns about the use of usually petroleum-based amine hardeners. 6 different amine hardeners were synthetized with varying amount of active hydrogens by using different aldehyde compounds. Four of them were purified and completely characterized allowing the authors to elaborate epoxy amine thermosets with both a petroleum-based and a bio-based epoxy compound. They successfully studied the properties of the new materials and compared them to existing similar epoxy-amine thermoset already reported in the literature.
The methodology concerning the synthesis of the amine hardener is not very well written and some doubts about the difference between IPTA1 and IPTA2 could be raised. Furthermore, the amine hardener resulting from the synthesis with vanillin and ,4-dihydrobenzaldehyde are not properly purified and characterized and are not studied in the rest of the work. Therefore, these two hardeners should not be mentioned in this manuscript.
The methodology of the elaboration of the epoxy amine materials and their characterization are clearly described and discussed.
Desirables:
Page 1, Line 30-33: The abbreviations list should be completed.
Page 6-7, Lines 163-180: ‘Synthesis of imines and its reduction’ should be renamed in a more general way such as ‘synthesis of amine hardeners’
The synthesis protocol is not very clear. This paragraph should refer to Figure 7.
Line 164: Authors should detail in which case they use MeOH or ‘H2O/2-MeTHF mixture (70 – 50)’ as a solvent. The ratio between the solvents should be written in a consistent way with figure 7 and should be corrected to ‘H2O-2-MeTHF mixture (70 – 30)’.
Line 166: ‘were solubilized in 50 mL of H2O or 2-MeTHF’ authors should precise here if the choice of the solvent is depending on the choice of the solvent for solubilisation for DYTEK® A and if so the case of MeOH as a solvent should be reported.
Line 177 : ‘Organic phase was washing with brine solution (400 mL). Then, organic phase was dry’ tense of verbs should be corrected.
Line 180: ‘All descriptions are considered as correspondence to one B and one A addition on IPA.’ : authors should rephrase.
Page 11, Line 261-262 : ‘according to Equation (1).’ The numbering of the equations is wrong, this equation should be rename ‘Equation (4).
Page 11, Line 266 – 270 Author should review the numbering of the equations as well as in text.
Page 11, Line 279‘applied to bio-based and nontoxic aromatic aldehydes with nontoxic diamines’: authors should write nontoxic diamine in singular as they use only DYTEK® A.
Page 12, Line 290-291 : ‘Vanillin and 3,4-dihydroxybenzaldehyde were also chosen due to their phenol moieties known to catalyze the epoxy-amine reaction and to improve the material adhesion.’ Hardener from Vanillin and 3,4-dihydrobenzaldehyde should not be reported in this manuscript. The characterizations of these compounds is not completely reported (No NMR of the amine compound from vanillin in supplementary information) and the hardener were not isolated from the reaction crude. Moreover, no thermoset materials were synthesized from these hardeners and the claimed catalytic effect of these compounds could not be studied.
Page 12, Line 302: ‘in order to favor monomer formation’ Did authors meant ‘mono-addition’? If so please change in texte.
Page 12, Line 304-308 The conditions of the one-pot synthesis of IPA-based amine are not clearly indicated. Is the one-pot synthesis concerning the addition of the aldehyde? Or the reduction step with NaBH4? This additional synthesis protocol should be clearly indicated in the experimental section as well as better description in text.
Page 13, Line 309 The structures d) and e) should be removed from the figure. Acronyms could appear directly on the structures.
Page 13, Line 312-313 : ‘The imine reduction allows obtaining full conversion of aldehyde during the imine synthesis step and then, full conversion of imine during the second step’ the imine reduction is not corresponding to the conversion of the aldehyde, the sentence should be corrected.
Page 13, Line 318-319 The sentence describing vanillin and 3,4-dihydroxybenzaldehyde use should be removed.
Page 14, Figure 8: Authors should add in title or in figure the corresponding monomer. They should also indicate by arrows the important bands to observe in the FTIR spectra.
Page 14, Line 333-335: ‘The IPA-based hardener synthesized in one-pot conditions (IPTA2) shows a similar spectrum aspect that IPTA1, with different integration values.’ The authors are referring to IPTA1 and IPTA 2 with different integration values, they should add the NMR spectra and the characterization of the IPTA 2 compound as it is used later as hardener.
Page 15, Line 344: ‘Thus, imine proton signals are shifted to 3.50 ppm.’ The authors are referring here to the proton in α of the iminium nitrogen, however in SI (P 4 part 2, a)) they indicate that the imine proton is the proton on the double bond carbon. Authors should homogenize.
Page 15, Figure 9: a) Authors should specify if it is the spectra of IPTA1 or IPTA2. These spectra were recorded with MeOD as solvent. However, in the experimental part as well as in SI, only NMR data in CDCl3 are reported and the chemical shifts reported in text are related to NMR in CDCl3 and not in MEOD. The authors should display full spectra (up to 12 ppm to see the absence of aldehyde proton) of the hardeners in CDCl3.
Page 16, Line 354-355: ‘The higher glass transition temperature (Tg) of IPTA2 compare to IPTA1 confirms the presence of more dimers in the IPTA2 hardener than in the IPTA1.’ The authors are confirming that they are more dimers in IPTA2 based on the Tg values. ‘Dimer’ should be replaced by ‘di-addition’ if the authors meant to discuss the di-addition of aldehyde on the diamine. Furthermore, the authors should bring prove before confirming in that way. The A and B addition ratio is clearly indicated, but the ratio between mono- and di-addition in the case of IPTA1 and IPTA2 is not studied. This information is missing in order to truly compare the properties of the materials in the following part.
Page 17, Line 368: ‘DSC values given in supporting information’ Authors should write ‘spectra’ instead of ‘values’. This should be changed in several places in text.
Page 18, Table 2: The optimal ratios for the synthesis of the epoxy-amine networks should be indicated in the table or in the text.
Page 19, Line 408: ‘Tg value twice more less than their comparable’: authors should rephrase.
References: the following references need to be updated with pages and volume : [15], [21], [28], [36]
Author Response
Reviewer 2
The manuscript submitted by Mora and co-workers describes the synthesis of new bio-based pluri-functional amine hardeners for the synthesis of epoxy thermosets. The authors clearly presented the objectives of the work and the global concerns about the use of usually petroleum-based amine hardeners. 6 different amine hardeners were synthetized with varying amount of active hydrogens by using different aldehyde compounds. Four of them were purified and completely characterized allowing the authors to elaborate epoxy amine thermosets with both a petroleum-based and a bio-based epoxy compound. They successfully studied the properties of the new materials and compared them to existing similar epoxy-amine thermoset already reported in the literature.
The methodology concerning the synthesis of the amine hardener is not very well written and some doubts about the difference between IPTA1 and IPTA2 could be raised. Furthermore, the amine hardener resulting from the synthesis with vanillin and ,4-dihydrobenzaldehyde are not properly purified and characterized and are not studied in the rest of the work. Therefore, these two hardeners should not be mentioned in this manuscript.
The methodology of the elaboration of the epoxy amine materials and their characterization are clearly described and discussed.
Desirables:
Page 1, Line 30-33: The abbreviations list should be completed. We have completed the list accordingly
Abbreviations: AHEW or HEW, amine hydrogen equivalent weight, BDA, benzyldiamine, BIA, benzylimineamine, DGEBA, diglycidyl ether of bisphenol A, DGEVA, diglycidyl ether of vanillyl alcohol, DMA, dynamic mechanical analysis, DSC, differential scanning calorimetry, EEW, epoxy equivalent weight, FDA, furfuryldiamine, FIA, furfurylimineamine, FTIR or IR, Fourier transform infrared, GC, gel content, IPA, isophtalaldehyde, IPDIDA, isophtalalatediiminediamine, IPTA, isophtalalatetetramine, MXDA, m‑xylylenediamine, NMR, nuclear magnetic resonance, PACM, 4,4’‑methylenebis(cyclohexylamine), ppm, parts per million, SI, swelling index, THF, tetrahydrofuran, TGA, thermogravimetric analysis.
Page 6-7, Lines 163-180: ‘Synthesis of imines and its reduction’ should be renamed in a more general way such as ‘synthesis of amine hardeners’. The synthesis protocol is not very clear. This paragraph should refer to Figure 7. We have modified the manuscript accordingly :
“Synthesis of amine hardeners (Figure 7)”
Line 164: Authors should detail in which case they use MeOH or ‘H2O/2-MeTHF mixture (70 – 50)’ as a solvent. The ratio between the solvents should be written in a consistent way with figure 7 and should be corrected to ‘H2O-2-MeTHF mixture (70 – 30)’. We understand the question of Reviewer. We have removed the procedure using MeOH. We have the same yields in the two case.
Line 166: ‘were solubilized in 50 mL of H2O or 2-MeTHF’ authors should precise here if the choice of the solvent is depending on the choice of the solvent for solubilisation for DYTEK® A and if so the case of MeOH as a solvent should be reported. Reviewer #2 is fully right and we apologize since we made a typo. In fact it was “MeOH or 2-MeTHF” not “H2O or 2-MeTHF”. However, we have removed the MeOH procedure.
Line 177 : ‘Organic phase was washing with brine solution (400 mL). Then, organic phase was dry’ tense of verbs should be corrected. We have corrected the manuscript accordingly
“Organic phase was washed with brine solution (400 mL). Then, organic phase was dried with MgSO4, filtered and solvent was removed under reduced pressure.”
Line 180: ‘All descriptions are considered as correspondence to one B and one A addition on IPA.’ : authors should rephrase. We have corrected manuscript accordingly.
“All the described structures correspond to the attack of each amine group from the diamine compound onto the dialdehyde (addition A or B displayed in Figure 7).”
Page 11, Line 261-262 : ‘according to Equation (1).’ The numbering of the equations is wrong, this equation should be rename ‘Equation (4). And Page 11, Line 266 – 270 Author should review the numbering of the equations as well as in text. Reviewer #2 is fully right. We have corrected numbering. Equations (1), (4) and (5) have been modified respectively in (4), (5) and (6).
Page 11, Line 279‘applied to bio-based and nontoxic aromatic aldehydes with nontoxic diamines’: authors should write nontoxic diamine in singular as they use only DYTEK® A. We have corrected the sentence accordingly
“with a non-toxic diamine”
Page 12, Line 290-291 : ‘Vanillin and 3,4-dihydroxybenzaldehyde were also chosen due to their phenol moieties known to catalyze the epoxy-amine reaction and to improve the material adhesion.’ Hardener from Vanillin and 3,4-dihydrobenzaldehyde should not be reported in this manuscript. The characterizations of these compounds is not completely reported (No NMR of the amine compound from vanillin in supplementary information) and the hardener were not isolated from the reaction crude. Moreover, no thermoset materials were synthesized from these hardeners and the claimed catalytic effect of these compounds could not be studied. All references to vanillin and 3,4-dihydroxybenzaldehyde have been removed from this paper, including scheme and supporting information. All “Supporting Information, parts” have been corrected.
Page 12, Line 302: ‘in order to favor monomer formation’ Did authors meant ‘mono-addition’? If so please change in texte. We understand the question of Reviewer. In fact, we intended to explain that the dropwise addition prevents the formation of oligomers which is desirable in order to favor only the formation of the targeted molecule (monomer). We have modified the manuscript to clarify explanation.
“Each aldehyde reactant was respectively added dropwise, in an excess of initial amine reactant, in order to avoid the oligomer formation.”
Page 12, Line 304-308 The conditions of the one-pot synthesis of IPA-based amine are not clearly indicated. Is the one-pot synthesis concerning the addition of the aldehyde? Or the reduction step with NaBH4? This additional synthesis protocol should be clearly indicated in the experimental section as well as better description in text. We understand the point raised by Reviewer. We have modified manuscript to improve clarity.
In experimental part:
“-In the case of one pot conditions for IPTA2 synthesis:
100 mL of H2O-2-MeTHF mixture (70 – 30) were added to DYTEK®A (17.3 g, 149 mmol, 10 equivalents) in a 250 mL round-bottom flask. Then, isophtalaldehyde (2 g, 14.9 mmol, 1 equivalent), were added. The reaction crude was stirred and heated until complete aldehyde conversion, at 110 °C. The solution was then cooled down to room temperature.
For each case, 2 equivalent of sodium borohydride was then added slowly to the theoretical amount of imine and solvent mixture”
In results and discussion part:
“Only IPA-based amine was also synthesized in one-pot conditions, by mixing entirely aldehyde at the same time as amine, to favor the dimerization,”
Page 13, Line 309 The structures d) and e) should be removed from the figure. Acronyms could appear directly on the structures. We have corrected the Figure accordingly.
Page 13, Line 312-313 : ‘The imine reduction allows obtaining full conversion of aldehyde during the imine synthesis step and then, full conversion of imine during the second step’ the imine reduction is not corresponding to the conversion of the aldehyde, the sentence should be corrected. We have corrected sentence accordingly:
“The reductive amination method allows obtaining full conversion of aldehyde during the imine synthesis step and then, full conversion of imine during the second step”
Page 13, Line 318-319 The sentence describing vanillin and 3,4-dihydroxybenzaldehyde use should be removed. All references to vanillin and 3,4-dihydroxybenzaldehyde have been removed from this paper.
Page 14, Figure 8: Authors should add in title or in figure the corresponding monomer. They should also indicate by arrows the important bands to observe in the FTIR spectra. We have modified the Figure 8 accordingly.
“Figure 8: imine signal disappearance of IPDIDA, the corresponding imine of IPTA amine monomer, followed by FTIR (up), and 1H NMR in CDCl3 (down)”
Page 14, Line 333-335: ‘The IPA-based hardener synthesized in one-pot conditions (IPTA2) shows a similar spectrum aspect that IPTA1, with different integration values.’ The authors are referring to IPTA1 and IPTA 2 with different integration values, they should add the NMR spectra and the characterization of the IPTA 2 compound as it is used later as hardener. Spectra have been added in the supporting information and changes have been made in the paper:
(spectra given in Supporting Information, parts 1, c).
Page 15, Line 344: ‘Thus, imine proton signals are shifted to 3.50 ppm.’ The authors are referring here to the proton in α of the iminium nitrogen, however in SI (P 4 part 2, a)) they indicate that the imine proton is the proton on the double bond carbon. Authors should homogenize. We understand the point raised by Reviewer #2 and we modified manuscript accordingly.
“However, before the reduction step, the α-CH2 of the imine signal is shifted to 3.50 ppm due to the conjugated system involving the imine bond.”
Page 15, Figure 9: a) Authors should specify if it is the spectra of IPTA1 or IPTA2. These spectra were recorded with MeOD as solvent. However, in the experimental part as well as in SI, only NMR data in CDCl3 are reported and the chemical shifts reported in text are related to NMR in CDCl3 and not in MEOD. The authors should display full spectra (up to 12 ppm to see the absence of aldehyde proton) of the hardeners in CDCl3. We understand the question of Reviewer #2. We have modified text and Figure 9 accordingly.
Page 16, Line 354-355: ‘The higher glass transition temperature (Tg) of IPTA2 compare to IPTA1 confirms the presence of more dimers in the IPTA2 hardener than in the IPTA1.’ The authors are confirming that they are more dimers in IPTA2 based on the Tg values. ‘Dimer’ should be replaced by ‘di-addition’ if the authors meant to discuss the di-addition of aldehyde on the diamine. Furthermore, the authors should bring prove before confirming in that way. The A and B addition ratio is clearly indicated, but the ratio between mono- and di-addition in the case of IPTA1 and IPTA2 is not studied. This information is missing in order to truly compare the properties of the materials in the following part. We understand the question of Reviewer #2. In this paper, all aldehydes were fully converted. In fact, we intended to discuss the formation of oligomers during the reaction. This oligomerization is induced by the addition of amines to IPA aldehyde and then by the second addition of the newly formed amine with another aldehyde. For instance, IPTA can react on one IPA molecule which has not yet reacted with DYTEK-A (competition reaction). In the case of dropwise addition, the formation of oligomers is prevented while in the case of one pot conditions, oligomers can occur.
Page 17, Line 368: ‘DSC values given in supporting information’ Authors should write ‘spectra’ instead of ‘values’. This should be changed in several places in text. We have replaced “DSC values” by “DSC thermograms” in the whole manuscript
Page 18, Table 2: The optimal ratios for the synthesis of the epoxy-amine networks should be indicated in the table or in the text. We have indicated the ratios in Tables in SI.
“DSC thermograms and optimal ratio given in supporting information, parts 6 and 7).”
Page 19, Line 408: ‘Tg value twice more less than their comparable’: authors should rephrase. We agree with Reviewer #2, this is not clear. We have corrected this sentence.
“Overall, fully bio-based thermosets exhibited a lower Tg value than the P‑material references, with a difference of 30 to 40 °C”
References: the following references need to be updated with pages and volume : [15], [21], [28], [36] Changes have been made in the paper: Carré, C.; Ecochard, Y.; Caillol, S.; Averous, L. From the synthesis of biobased cyclic carbonate to polyhydroxyurethanes: a promising route towards renewable NonIsocyanate Polyurethanes. ChemSusChem 2019, 12, 3410–3430. Mora, A.-S.; Tayouo, R.; Boutevin, B.; David, G.; Caillol, S. Vanillin-derived amines for bio-based thermosets. Green Chem. 2018, 20, 4075–4084. Bornadel, A.; Bisagni, S.; Pushpanath, A.; Montgomery, S.L.; Turner, N.J.; Dominguez, B. Technical Considerations for Scale-Up of Imine-Reductase-Catalyzed Reductive Amination: A Case Study. Org. Process Res. Dev. 2019, 23, 1262-1268. Wang, H.; Gao, Z.; Wang, X.; Wei, R.; Zhang, J.; Shi, F. Precise regulation of selectivity of supported nano-Pd catalysts using polysiloxane coatings with tunable surface wettability. Chem. Commun. 2019, 55, 8305–8308.
Round 2
Reviewer 1 Report
The authors have addressed all concerns of this reviewer in their revised version, which significantly improved the quality of the manuscript.
Reviewer 2 Report
The authors answered all questions and corrected the manuscript in an adequate way. I recommend the paper to be considered for publication in its present form.